# Anti-Herpetic, Anti-Dengue and Antineoplastic Activities of Simple and Heterocycle-Fused Derivatives of Terpenyl-1,4-Naphthoquinone and 1,4-Anthraquinone [note 1]

**DOI:** 10.3390/molecules24071279

**Published:** 2019-04-02

**Authors:** Vicky C. Roa-Linares, Yaneth Miranda-Brand, Verónica Tangarife-Castaño, Rodrigo Ochoa, Pablo A. García, Mª Ángeles Castro, Liliana Betancur-Galvis, Arturo San Feliciano

**Affiliations:** 1Group of Investigative Dermatology, Institute of Medical Research, Faculty of Medicine, University of Antioquia, Medellin 050010, Colombia; vicky.roa@udea.edu.co (V.C.R.-L.); yaneth.miranda@udea.edu.co (Y.M.-B.); verotanga@gmail.com (V.T.-C.); 2Programa de Estudio y Control de Enfermedades Tropicales PECET, Facultad de Medicina, University of Antioquia, Medellín 050010, Colombia; rodrigo.ochoa@udea.edu.co; 3Departamento de Ciencias Farmacéuticas, Área de Química Farmacéutica, Facultad de Farmacia, CIETUS, IBSAL. Campus Miguel de Unamuno, University of Salamanca, 37007-Salamanca, Spain; pabloagg@usal.es (P.A.G.); artsf@usal.es (A.S.F.)

**Keywords:** terpenylquinones, naphthoquinones, anthraquinones, antiviral activity, herpesvirus, dengue virus, cytotoxicity

## Abstract

Quinones are secondary metabolites of higher plants associated with many biological activities, including antiviral effects and cytotoxicity. In this study, the anti-herpetic and anti-dengue evaluation of 27 terpenyl-1,4-naphthoquinone (NQ), 1,4-anthraquinone (AQ) and heterocycle-fused quinone (HetQ) derivatives was done in vitro against Human Herpesvirus (HHV) type 1 and 2, and Dengue virus serotype 2 (DENV-2). The cytotoxicity on HeLa and Jurkat tumor cell lines was also tested. Using plaque forming unit assays, cell viability assays and molecular docking, we found that NQ **4** was the best antiviral compound, while AQ **11** was the most active and selective molecule on the tested tumor cells. NQ **4** showed a fair antiviral activity against Herpesviruses (EC_50_: <0.4 µg/mL, <1.28 µM) and DENV-2 (1.6 µg/mL, 5.1 µM) on pre-infective stages. Additionally, NQ **4** disrupted the viral attachment of HHV-1 to Vero cells (EC_50_: 0.12 µg/mL, 0.38 µM) with a very high selectivity index (SI = 1728). The in silico analysis predicted that this quinone could bind to the prefusion form of the E glycoprotein of DENV-2. These findings demonstrate that NQ **4** is a potent and highly selective antiviral compound, while suggesting its ability to prevent Herpes and Dengue infections. Additionally, AQ **11** can be considered of interest as a leader for the design of new anticancer agents.

## 1. Introduction

In recent years, the treatment and control of some viral agents has become a major challenge for the pharmaceutical industry due to their pathophysiological features and development of drug-resistance. Among these, Human Herpesviruses type 1 and 2 (HHV-1 and HHV-2), associated with cold sore and herpes genitalis respectively [1], are neurotropic and neuro-invasive double-stranded DNA (dsDNA) viruses with the ability to induce persistent infection during the lifetime of the host, often in latent form with sporadic reactivation episodes induced by environmental or immunological conditions [2]. Therapy with nucleoside analogues such as acyclovir, whose mechanism of action is the inhibition of viral DNA polymerase, has been widely applied for the treatment of herpesvirus infections [3]. Nonetheless, drug-resistant strains are an emerging concern, especially in immunocompromised patients who are treated with these antivirals for a long time [3,4,5].

Dengue disease, caused by Dengue virus (DENV), is currently the most prevalent mosquito-borne viral infection around the world, causing approximately between 50 and 200 million symptomatic cases, and 20,000 deaths per year [6]; remarkably, there are no drugs available for its treatment. Several efforts have been done to counter DENV spread via the biological and chemical mosquito control and the recent implementation of a licensed vaccine. Nevertheless, these strategies can induce some side effects, such as selection of resistant vectors [7,8] and increased risk of illness progression to more severe grades [9]. Therefore, the research of antiviral molecules to treat this disease is paramount. Like other positive-sense single-stranded RNA viruses, DENV has high evolution rates; therefore, the development of highly potent and selective antivirals can face a number of difficulties.

Additionally, cancer is another health problem, the second leading cause of death globally. Cervix cancer represents the fourth most frequent and deadly type of cancer in women worldwide, with an estimate of 570,000 new cases in 2018, contributing 3.6% of all female cancer deaths [10]. Meanwhile, leukemia is the fifteenth most commonly occurring cancer and the eleventh cause of cancer death on both sexes worldwide [10]. Despite the wide variety of current chemotherapeutic drugs for cancer treatment, they cause many side effects, including damage of heart, lungs, kidneys and brain. Furthermore, some treatments have a limited anti-cancer activity [11].

Seeking for potential effective and selective antiviral and anti-tumor treatments, substances obtained from natural products provide unlimited opportunities for new drugs due to their availability and chemical diversity [12,13]. In this regard, quinones are phenolic-related secondary metabolites that exhibit diverse pharmacological properties, including antiviral, antimicrobial and anti-inflammatory activities, and represent a class of important anticancer agents [14].

Several reports have demonstrated the antiviral activity of some naphthoquinones (naphthazarins and shikonin) and 9,10-anthraquinones (emodin and doxorubicin derivatives) [15,16,17]. Likewise, it was reported on the cytotoxicity of some naphthoquinones against tumor cells [18,19]. On these subjects of study, we also previously reported that some terpenyl-1,4-naphthoquinone (NQ), 1,4-anthraquinone (AQ) and heterocycle-fused quinone (HetQ) derivatives showed cytotoxicity on several types of tumor cell lines and moderate activity against herpesviruses, yeasts and filamentous fungi [20,21].

The mechanisms of action of quinones are mainly related to their ability to inhibit electron transport and to uncouple oxidative phosphorylation. They can also act as intercalating agents into the DNA double helix, as bioreductive alkylators of biomolecules, and as inducers of reactive oxygen species (ROS) [14]. In particular, the antifungal activity of quinones has been related to ROS generation.

Aiming to discover more potent and selective antiviral and anticancer molecules, the biological evaluation of an additional set of NQs, AQs and HetQs has been done. In this study, we have evaluated the anti-herpetic, anti-dengue and antineoplastic cytotoxicity of 27 compounds.

## 2. Results and Discussion

### 2.1. Chemistry

Quinones and their derivatives were prepared by the initial Diels–Alder cycloaddition reaction between myrcene and *p*-benzoquinones followed by appropriate transformations, as previously described by us [20,21]. The compounds included in this study were selected as representative members of three families of terpenyl-1,4-NQs (**1**–**9**), 1,4-AQs (**10**–**18**) and HetQs (**19**–**27**), and their structures are represented in Table 1 and Figure 1.

### 2.2. Biological Evaluation

#### 2.2.1. Antiviral Activity

The in vitro antiviral evaluation of the 27 substances against Human Herpesvirus type 1 (HHV-1) and 2 (HHV-2) was made on infected Vero cells, using the end-point titration technique (EPTT) [22]. The compounds that showed a fair reduction of viral titer at concentrations ≤50 μg/mL, after 48 h, were considered active. Table 2 shows the reduction values of viral titer (*Rf*) and the antiviral activity (µg/mL) of those NQs, AQs and HetQs active against at least one virus serotype.

According to the estimate of Vlietinck et al. [23], a purified natural molecule is considered to have a relevant or moderate antiviral activity when the reduction factor (*Rf*) of viral titer is ≥1 × 10^3^ or 1 × 10^2^, respectively. In this study, we define the *Rf* of 1 × 10^1^ or ≥1 × 10^2^ for mild or moderately active substances, respectively.

As shown in Table 2, a total of thirteen compounds were active against HHV-1 or HHV-2. The NQ **7**, and the AQs **15**–**17** showed mild activity (*Rf* = 1 × 10^1^) against HHV-1 at a concentration of 50 µg/mL, while the AQ **18** showed the same *Rf* value at a concentration of 25 µg/mL. Against HHV-2, NQ **2** was mildly active at 25 µg/mL and AQ **13** at 12.5 µg/mL. NQs **2**, **4**, **6**, **8** and AQ **10** showed moderate effect against HHV-1, with a *Rf* of 1 × 10^2^ at 6.25 µg/mL, except for NQ **8**, that attained the same *Rf* (1 × 10^2^) at the concentration of 25 µg/mL. A moderate antiviral activity (*Rf* = 1 × 10^2^) was also found for NQs **3**, **4** and for AQ **12** against HHV-2 at the concentrations of 6.25, 12.5 and 25 µg/mL, respectively.

These results reveal that, in general, 1,4-NQs are more potent anti-herpetic substances than 1,4-AQs, and that NQ **4** is the only quinone that showed moderate anti-herpetic activity against both HHV-1 and HHV-2 serotypes, suggesting a wide spectrum of antiviral activity for this molecule. From the chemical point of view, the presence of one or two chlorine atoms in the quinone ring, such as in NQs **2**, **4** and **6**, and AQs **10** and **12**, contributes to enhance the antiviral activity. This feature probably correlates with the broad spectrum of activity of these substances, also in accordance with our previous study, in which the chlorinated quinones also showed significant antifungal activity [20].

In general, we note that the molecules tested were effective mainly against HHV-1 (Table 2). This fact should be related to the own molecular nature of the compounds and to the dose of infectious virus employed in our assays. Nonetheless, it is important to note that each HHV serotype uses different cellular receptors to attach and enter the host. Notably, HHV-1 has more receptors available for infection than HHV-2 [24]. This feature can be a disadvantage for HHV-1, considering that there are also a greater number of binding sites accessible for the antiviral agents, thus leading to a possible disruption of the viral entry and the subsequent replication steps.

To define the stage of the viral replicative cycle where NQ **4** exerts its action, simultaneous and post-infection treatments against HHV-1 and HHV-2 were performed. Additionally, this experiment was also extended to DENV-2 to prove the potential antiviral broad-spectrum of NQ **4** (Figure 2). First, we determined the inhibitory concentration 50% (IC_50_) for the compound and drug positive controls on infected cells and their effects on non-tumoral Vero cells, to define the concentration of evaluation and to calculate the antiviral selectivity index (SI). Then, the concentration of NQ **4** that reduced the number of viral plaques by 50% (EC_50_) was determined from the dose-response curves. In this study, we have considered that a molecule has interesting antiviral selectivity for a SI (IC_50_/EC_50_) value > 10.

On Vero cells, NQ **4** showed an IC_50_ value > 200 µg/mL, and the antiviral controls: dextran sulfate (DS), heparin (H), acyclovir (A) and ribavirin (R) showed IC_50_ values > 400 µg/mL (*r*^2^ = 0.84, data not shown). In the simultaneous treatment, NQ **4** showed anti-herpetic activity in all concentrations tested (0.4–3.1 µg/mL) in a significant manner (*p* value < 0.001) for both HHV-1 (Figure 2A) and HHV-2 (Figure 2B), thus, it was not possible to calculate dose-response curves and antiviral SI. As expected, dextran sulfate positive control was active against HHV-1 and HHV-2 at 5 µg/mL. Against DENV-2 (Figure 2C), NQ **4** was evaluated at lower concentrations (0.4–1.6 µg/mL) that against HHV serotypes, showing significant antiviral activity only at 1.6 µg/mL (*p* value < 0.001). This effect was comparable to that displayed by the positive control heparin (10 µg/mL). In post-infective stages, NQ **4** showed no activity, neither against HHV-1, HHV-2 nor DENV-2. However, positive controls acyclovir and ribavirin employed against HHV serotypes and DENV-2 respectively, were active. These results suggest that NQ **4** produces its antiviral effect during early stages of the infectious cycle, i.e., on the attachment and/or the viral entry.

To prove this hypothesis, we evaluated if NQ **4** exerts its effect on attachment, when the reversible interaction between viral glycoproteins and cellular receptors occurs, or on the viral entry, when membrane fusion and virus internalization happen. Taking into account the high activity of this molecule in simultaneous treatment, lower concentrations (0.8–0.1 µg/mL) of NQ **4** were employed. This assay was performed only against HHV-1 considering, as mentioned above, that this serotype uses more cellular receptors for attachment and entry.

Results showed that NQ **4** reduces significantly (*** *p* < 0.001) the HHV-1 attachment on Vero cells (EC_50_ = 0.12 µg/mL and SI = 1728) compared to DS and DMSO controls. Meanwhile, the effect of this compound on viral entry was not significant (Figure 3).

Some reports indicate that HHV-1 make the initial contact with the host cells by binding to glycosaminoglycan receptors, such as heparan sulfate; this interaction is reversible but necessary for the virus’s location on the cell surface and thus, to allow the binding of viral ligands to specific receptors [25]. Additionally, it has been described that exist different ways of HHV entry to cells, including low pH-dependent or independent endocytosis and fusion at the plasma membrane. Even though these viral entry pathways are cell-type dependent, it should be noted that glycoproteins gB, gD and gH/gL are required for both entry pathways [26].

In recent years, antiviral activity has been described for certain quinones and other structurally related molecules with therapeutic potential. NQs and AQs have shown antiviral activities against DNA and RNA viruses by inhibition of viral entry, replication and genome transcription, as well as affecting the function of important enzymes for the viral replicative cycle [27,28,29,30,31,32,33]. Emodin, a 9,10-anthraquinone isolated from the roots of *Rheum tanguticum*, showed in vitro and in vivo anti-herpetic activity affecting viral replication, i.e., in late stages of infectious cycle [15]. Likewise, other antiviral mechanisms have been described for this quinone, including inhibition of UL12 [34], a protein related to the uncoating and DNA processing, and the inhibition of enzymes involved in viral proteins phosphorylation such as casein kinase [35]. Moreover, denbinobin, a phenanthrenequinone, has been reported as dual inhibitor of the HIV-1 LTR promoter and the transcription factor NF-kB affecting the viral transcription [36].

Our results demonstrate that NQ **4** acts selectively on early infection stages of the HHV-1 strain, specifically on viral attachment. This suggests that this NQ should mainly disrupt the interaction between the viral glycoproteins and glycosaminoglycan receptors, therefore, the viral entry would be partially affected (Figure 3). This antiviral approach is more attractive because these steps are mandatory for successful viral infection.

#### 2.2.2. Molecular Docking Study of NQ **4** on DENV-2 

To reinforce the hypothetical mode of action of NQ **4** on DENV-2 attachment or entry, a molecular docking analysis of this compound with the β-OG (*n*-octyl-β-d-glucoside) binding cleft of envelope protein (ENV) pre-fusion form (PDB:1OKE) was run. Based on the co-crystallized complex, we determined the potential pocket site for the discovery of small-molecule fusion inhibitors [37]. On this pocket, several amino acids have been reported to be critical for membrane fusion during virus entry, among which are Thr48, Glu49, Ala50, Lys51 and Gln52 [37,38]. Our studies show that NQ **4** fits its aliphatic substructure into the β-OG binding cleft of the dengue virus E glycoprotein dimer with a predicted score of −7.7 kcal/mol. The molecule forms a key hydrogen bond with Ala50 and Gln200, and a set of hydrophobic interactions within the cleft with Glu49, Leu135, Phe193, Leu198, Leu207 and Ile270 (Figure 4A). According to the predicted mode of action, the enthalpic contribution is mostly governed by hydrophobic interactions that are positively induced by the preferred opposite orientation of the two chlorine atoms.

To understand the impact of such interactions, we also docked the anthracycline antibiotic doxorubicin. This drug was evaluated by Kaptein et al. [16], together with a doxorubicin derivate (SA-17) that has a squaric acid amide ester moiety at the carbohydrate group. Both were active against DENV-2 at low concentrations during the very early stages of the viral replication cycle (i.e., virus attachment and/or virus entry). During SA-17 treatment, time-of-addition studies revealed that at concentration of 10 µg/mL (15 µM), the drug attained 98% and 42% inhibition of virus replication when was added at first 0 and 2 hours of infection, respectively. Additionally, this compound failed to efficiently inhibit viral replication when was added between 4 and 12 h post-infection and did not inhibit DENV-2 RNA replication. Using the same β-OG binding site of ENV protein, Kaptein et al. [16] reported molecular docking studies for SA-17, that formed hydrogen bonds with the amino acids Ala50, Tyr137 and Gln200, and had hydrophobic contacts with Thr48, Pro53, Lys128, Leu135, Phe193, Leu198, Ala205, Ile270, Gln271, Thr280 and Gly281, some of which are critical in membrane fusion during virus entry [16]. In our case, we found that doxorubicin forms hydrogen bonds with Glu26 and His27 and displays π-stacking interactions with Phe279 (Figure 4B). The molecule fits part of its structure into the β-OG binding cleft of the dengue virus E glycoprotein dimer with a similar but lower score (−6.5 kcal/mol) than NQ **4**. However, no interactions were formed with any of those amino acids found for NQ **4**, or those reported for SA-17 [16]. Considering this information, it is possible to propose that NQ **4** acts in a similar way as SA-17, probably affecting DENV-2 entry, rather than its attachment as we found for HHV-1.

The antiviral activity of several quinones against herpesviruses in late stages of infection has been reported [29,30,31]. In this study, we have found a different kind of antiviral effect of quinones, showing that NQ **4** exerts anti-HHV-1 activity in early stages, specifically avoiding viral attachment. This compound possibly disrupts the entry of DENV-2. Some quinone derivatives may affect dramatically phospholipidic membranes and may be responsible for remarkable changes in their physical and biological properties. These alterations might consist in changes of lipid/water interface in negatively charged phospholipids and disruptions on the core of lipid bilayer [39]. This report may be consistent with our findings, considering that the lipidic membrane reconfiguration caused by this type of quinones may affect the initial interaction virus-cellular receptor, and the subsequent viral entry to the host cell. Indeed, on the basis of the predicted logP (clogP) values 3.47 (6.487), and the low tPSA (total Polar Surface Area) value of 44.6 estimated through the ChemDraw algorithms [40] for NQ **4**, its membrane affinity appears fairly supported. However, additional experiments are necessary to confirm or refuse this mechanistic hypothesis.

#### 2.2.3. Cytotoxicity

The 27 NQs, AQs and HetQs were tested to evaluate their in vitro cytotoxicity on tumor (HeLa and Jurkat) and non-tumor (Vero) cells. The common anticancer drug doxorubicin was included in the assays as reference. As seen in Table 3, most of the molecules were cytotoxic for at least one cancer cell line, and from the results obtained, some general considerations can be made for the three groups of quinones tested.

To analyze the results, we consider that compounds with IC_50_ ≤ 10 μM have good cytotoxicity against neoplastic cells, while those with IC_50_ values between 10 and 50 μM have a moderate cytotoxicity, and IC_50_ values ≥ 50 μM correspond to low to null activity. Additionally, several recent studies have defined that the selectivity index (SI) value of 14.3 with respect to HepG2 cells is considered as indicative of potential therapeutic use for anticancer agents [41]. Such a SI value serves as criterion in this work to define a selective anticancer substance.

As a general observation, AQs were more cytotoxic than NQs, and both showed better results than HetQs in accordance with our previous studies [20,21]. Among the NQs, the presence of halogens on the quinone ring improved the cytotoxicity on both cancerous and normal cells, while in the AQ group an arylamino substituent induced higher selectivity leading to the best neoplastic cytotoxicity with low toxicity for normal cells. Within the group of HetQs, only the imidazole-fused HetQ **23** and the pyrazine-fused HetQ **25** can be considered moderately cytotoxic on one or both HeLa and Jurkat cells. Therefore, HetQ **23** showed IC_50_ values of 15.5 and 13.5 μM on both cell lines respectively, but with low SI values, whereas HetQ **25**, which showed a moderate cytotoxicity on Jurkat cells (13.8 μM), attained a higher selectivity (SI = 20).

The quinone with the highest in vitro cytotoxicity on HeLa cells was AQ **11**, bearing a *p*-methoxyphenylamino substituent. It showed the lowest IC_50_ value of 10 nM and the best SI value near 14 × 10^3^ with respect to Vero cells, much better than that of the reference drug doxorubicin. In addition, AQs **16** and **17** which were cytotoxic for HeLa cells with IC_50_ values of 5.6 µM and 7.5 µM, displayed SI values of 32 and 16, respectively. A SI of 14 was also observed for the dichlorinated NQ **4**, though it was in the range of low cytotoxicity (IC_50_ = 46.6 µM). On Jurkat cells, again AQ **11** showed a high cytotoxicity (1.4 µM) and the relevant SI value of 168, followed by NQs **2** and **3** and AQs **13** and **17,** which showed IC_50_ values of 6.2, 1.6, 6.3 and 9.6 µM, respectively; though with moderate to low SI values of 15, 8, 11 and 12, respectively. In addition, AQ **11** and AQ **17** were quinones with fair cytotoxicity on HeLa and Jurkat cells at lower concentrations. Finally, according to their SI values, the AQs **11**, **16** and **17** were the most selective compounds on both types of cancer cells, unlike NQ derivatives.

It must be noted that in addition to the very high selectivity towards cancer cells, AQ **11** also showed an interesting selectivity depending on the type of cancer, resulting some 800 times more potent against HeLa than against Jurkat cells (Figure 5). This fact also defines a qualitative difference with respect to doxorubicin, which inversely resulted much less selective, being only some ten times more cytotoxic for Jurkat than for HeLa cells. 

Aiming to obtain further validation of AQ **11**, its structure was submitted online to predictive screenings to get data on its druggability potential. Therefore, on examination under the prediction algorithms of Molinspiration virtual screening engine v2018.03 [42], AQ **11** was qualified as potential kinase inhibitor (score: 0.29) and as potential nuclear receptor ligand (0.23), that is recognizing its probable intrinsic bioactivity. Further examination under the Osiris property explorer, in addition to physicochemical data like MW (361.44) and clogP (4.14) values within those permitted by the Lipinski Rule of Five, revealed that AQ **11** was potentially exempt of carcinogenic, mutagenic, irritant and on the reproductive cycle effects; that is devoid of main adverse effects [43]. All these experimental facts and calculated or predicted data support the selection of AQ **11** for carrying out mechanistic studies, target definition and structure optimization, oriented to configure preclinical toxicity and efficacy assays.

Among the mechanisms of action of quinones in different tumor types, it has been reported the decrease in mitochondrial membrane potential through ROS-mediated pathway [44,45], as the G2/M-phase arrest through the down-regulating of G2/M regulatory proteins cyclin B1 and Cdc25B [46], or the DNA damage induction through double-strand breaks by inhibition of topoisomerase II and the glutathione depletion [47]. Additionally, the quinone moiety is present in some clinically useful anticancer agents as mitomycin C or doxorubicin. Mitomycin C is used in the treatment of gastrointestinal tumors acting as a double-strand DNA alkylating agent [48], and doxorubicin is commonly used in cancer treatment, including breast, lung and gastric carcinomas among other, acting as DNA intercalating agent and generating free radicals that damage cellular membranes, DNA and proteins [49]. All these facts and these examples of clinically useful drugs demonstrate the potential of quinone derivatives as antitumor agents. Furthermore, our results suggest that cell death induced by AQ **11**, AQ **16** and AQ **17** could involve one or more of those above-mentioned mechanisms of action or even different ones. Consequently, additional studies are required to establish the proper cytotoxic mechanism for these AQs.

## 3. Materials and Methods

### 3.1. Chemistry

NQs **1, 2, 3, 5** and **8** were obtained by previously described procedures [50]. AQs **10** and **12** were also obtained as described before [51]. NQs **4, 6, 7, 9** and AQs **11, 13** and **14**–**17** and **18** were obtained as described previously [20]. HetQs **19**–**21** and **22**–**27** were obtained as described previously [21].

### 3.2. Biological Evaluation

#### 3.2.1. Samples, Cells and Viruses

Stock solutions of the compounds were prepared in dimethyl sulfoxide (DMSO, Sigma, Cream Ridge, NJ, USA) and frozen at −70 °C until use. The concentration of DMSO in biological assays was 0.05%. Cell controls with DMSO at 0.05% were used.

For antiviral assays, Vero (African green monkey kidney-*Cercopithecus aethiops*, ATCC CCL-81) and BHK-21 (Baby hamster kidney fibroblasts-*Mesocricetus auratus*, ATCC CCL-10) cell lines were maintained in Dulbecco’s Modified Eagle’s Medium (DMEM) supplemented with 5% fetal bovine serum (FBS) and incubated at 37 °C in humidified 5% CO_2_ atmosphere. C6/36HT *(Aedes albopictus*-ATCC CRL-1660) cell line was maintained in DMEM supplemented with 10% FBS and incubated at 34 °C in humidified 5% CO_2_ atmosphere. For cytotoxicity evaluation, cell lines of human cervix epithelioid adenocarcinoma cells (HeLa, ATCC CCL-2) and acute T cell leukemia (Jurkat, ATCC TIB-152) were used, as well as, the non-tumor cell line Vero. Vero and HeLa cells were grown in DMEM supplemented. Jurkat cells were maintained in RPMI-1640 medium (supplemented with 10% FBS) at 37 °C in humidified 5% CO_2_ atmosphere. In all cases, maintenance media were supplemented with inactivated FBS, 100 units/mL of penicillin and 100 µg/mL of streptomycin.

Human Herpesvirus 1 (HHV-1, CDC Atlanta acyclovir-sensitive strain) and Human Herpesvirus 2 (HHV-2, VR-734-G acyclovir-sensitive strain) obtained from the Center for Disease Control (Atlanta, GA, USA), were amplified in Vero cells. Virus stocks were titrated in Vero cells by plaque assay and expressed as plaque forming units (PFU/mL). Dengue virus type 2 (DENV-2, New Guinea strain) was donated by Maria Elena Peñaranda and Eva Harris (Sustainable Sciences Institute and the University of California at Berkeley). DENV-2 was amplified in C6/36HT cell line and titrated in BHK-21 cells following our laboratory conditions.

#### 3.2.2. Screening for Anti-Herpetic Activity

The antiviral activity of molecules against 1 and 10 Cell Culture Infectious Dose 50% (10TCID_50_) of HHV-1 and HHV-2, respectively, was determined using end-point titration technique (EPTT) [22]. Vero cells grown in 96-well plates at a density of 2.0 × 10^4^ cells/well were incubated at 37 °C in 5% CO_2_ atmosphere until constitute 80% of cell monolayer. Then, viral suspensions of HHV-1 or HHV-2 with concentrations of 6.25 µg/mL to 50 µg/mL of compounds were performed in DMEM supplemented with 2% FBS containing 1% and 0.5% carboxymethylcellulose (CMC) for HHV-1 and HHV-2, respectively. Mixture was incubated during 15 minutes at room temperature and was added to the cell monolayer. After 48 h of incubation at 37 °C (5% CO_2_), the cytopathic effect was examined, the microplates were fixed with 3.5% formaldehyde and stained with 0.2% crystal violet. Two independent experiments by quadruplicate for each viral serotype and each were carried out. Positive controls, such as dextran sulfate (DS) and acyclovir (A) for early stages and late stages of infection, respectively, were included.

#### 3.2.3. Simultaneous and Post-Infection Treatment on HHV-1, HHV-2 and DENV-2

The potential antiviral mechanism of NQ **4** was evaluated by the plaque reduction assay as previously described by us [52]. Vero cell monolayers grown in 24-well plates were infected with 100 PFU/well of each virus. For simultaneous treatment, the compound and the virus were added simultaneously to the cell monolayers and incubated for 1 h at 37 °C (5% CO_2_). Then, cells were washed with phosphate buffered saline (PBS, pH = 7.0) and CMC at 1% and 0.75% was added for HHV-1 and HHV-2, respectively. In post-infection treatment, virus was added to cell monolayer and incubated for 1 h at 37 °C (5% CO_2_). After incubation, washing was performed with PBS and molecule previously prepared in CMC 1% and 0.75% for HHV-1 and HHV-2 respectively, was added. In both treatments, NQ **4** was prepared at concentrations from 0.4 µg/mL to 3.1 µg/mL and the cell monolayers allowed to incubate for 72 h. Subsequently, cells were fixed and stained with a solution of 3.5% formaldehyde with 0.2% crystal violet, and the viral plaques were counted. Dextran sulfate (DS) was included as positive control in simultaneous assay and acyclovir (A) was the positive control in after infection treatment.

Against DENV, the effect of NQ **4** either simultaneously or post-infection was also evaluated by the plaque reduction assay as previously described. Concentrations employed for this assay were from 0.4 µg/mL to 1.6 µg/mL. In this test, BHK-21 cell monolayers grown in 24-well plates were infected with 100 PFU/well of DENV-2 and treatments were performed in the same way to describe for Herpesviruses. Moreover, the monolayers were incubated for 6 days, fixed and stained with a solution of 3.5% formaldehyde with 0.2% crystal violet, and the viral plaques were counted. Heparin (H) was included as a positive control in the simultaneous assay, and ribavirin (R) was the positive control in after infection treatment.

#### 3.2.4. Evaluation of Anti-HHV-1 Mechanism of Action in Pre-Infective Stages

The effect of NQ **4** in the initial phases of the HHV-1 viral replication was done as previously described by Cardozo et al., 2011 [53]. In the attachment assay, pre-chilled (1 h at 4 °C) Vero cell monolayers were exposed to viruses (100 PFU/well) in presence or absence of the compound and were incubated for 2 h at 4 °C. After incubation, the substance and unbounded viruses were removed with cold PBS; cells were overlaid with CMC 1% and incubated for 72 h at 37 °C (5% CO_2_). In the entry assay, pre-chilled cells were infected with viruses (100 PFU/well) and incubated for 2 h at 4 °C. After this time, the unbound viruses were removed with cold PBS; cells were treated with different concentrations of pre-warmed compound, and then incubated for 1 h at 37 °C. Unabsorbed viruses were inactivated using citrate buffer (pH = 3.0) and then cells were washed with PBS, overlaid with CMC 1%, and incubated for 72 h at 37 °C (5% CO_2_). In both assays, further procedures were the same mentioned previously for the plaque reduction assay and dextran sulfate (DS) was included as positive control.

In these tests, the evaluated NQ **4** concentration range was from 0.1 to 0.8 µg/mL and the effective concentration fifty (EC_50_), the concentration that reduces the 50% of plaque forming units, was determined from dose-effect curves by linear regression methods for each compound. EC_50_ values were expressed as the mean ± SEM (standard error of the mean) of at least four dilutions by quadruplicate. Additionally, to determinate if NQ **4** was selective for infected cells rather than uninfected cells, it was calculated the antiviral selectivity index (SI), defined as the ratio between the inhibitory concentration 50 (IC_50_) on Vero cells and the EC_50_ for each virus.

#### 3.2.5. Molecular Docking with DENV-2 Prefusion Envelope Protein

Parametrization of ligands (NQ **4** and doxorubicin) and DENV-2 prefusion envelope protein (PDB:1OKE) was done using the AutoDock Tools suite [54]. Hydrogen atoms were added to the polar side chains and partial charges were calculated through the Gasteiger methodology. Then, a grid box was delimited in a binding site previously reported with some studied inhibitors [16]. Molecular docking was run with a modified version of AutoDock Vina that includes a scoring function parameterized also for halogen interactions [55]. We used an exhaustiveness (number of internal repetitions) of 20 for each protein-compound pair. The interactions (hydrogen bonds and hydrophobic interactions) and the predicted free energy scores in kcal/mol were obtained. Visualization of the docking results was generated using the Discovery Studio package.

#### 3.2.6. Cytotoxicity Assay

The in vitro cytotoxicity evaluation of quinones was performed using the 3-(4,5 dimethylthiazol-2-yl)-2,5-diphenyltetrazolium bromide (MTT, Sigma, Cream Ridge, NJ, USA) assay as described by Betancur-Galvis et al. 2002 [22]. Briefly, Vero and HeLa cell lines were seeded at 2.0 × 10^3^ cells per well of 96-well plates in DMEM supplemented with 10% of inactivated FBS, and were incubated for 24 h at 37 °C, 5% CO_2_. Then, each diluted molecule was added to the cells and incubated for further 48 h at 37 °C, 5% CO_2_. Jurkat cell line at 3 × 10^3^ cells per well in a 96-well round-bottomed plate and diluted substances in RPMI-1640 medium (Sigma) supplemented with 10% FBS were plated simultaneously. After 48 h of treatment at 37 °C, 5% CO_2_, the media was carefully removed and 28 µL of MTT solution (4 mg/mL) was added to each well, and the plates were incubated for 2 h at 37 °C, 5% CO_2_. The DMSO was then added to dissolve the formed formazan crystals and absorbance was determined spectrophotometrically at 570 nm.

The minimal dilution of compound that caused 50% inhibition of the cells (IC_50_) was calculated by linear regression analysis of the dose-response curves generated from the absorbance data with the statistical GraphPad Prisma 5.0. IC_50_ values were expressed as the mean ± standard deviation (M ± SD) of two independent experiments done in quadruplicate.

To define which molecules were more selective against cancerous cells, the selectivity index (SI), defined as Vero IC_50_ over HeLa or Jurkat IC_50_ values, was calculated.

#### 3.2.7. Statistical Analysis

Statistical analyses were performed using the statistical software GraphPad Prism® v. 5.0 (GraphPad Software Inc., La Jolla, CA, USA). In all cases, *p* value < 0.05 was statistically significant.

## 4. Conclusions

Currently, it is necessary to discover new and better antivirals with novel mechanisms of action for the treatment of Human Herpesvirus type 1 and 2 infections, mainly for the treatment of immunocompromised and transplanted patients, considering the continuous emergence of HHV acyclovir-resistant strains. Moreover, to discover medicines for Dengue disease is imperative, taking into account the impact of this disease on public health. 

In our study, the naphthoquinone NQ **4** has shown important anti-herpetic (EC_50_: <0.4 µg/mL, <1.28 µM) and anti-dengue (1.6 µg/mL, 5.1 µM) activities on early infection stages, mainly in the initial formation of complexes between the viral glycoprotein and the host cell surface receptors, thereby preventing all events related to fusion, and subsequent viral genome replication, and production of new virions. Additionally, NQ **4** disrupted the viral attachment of HHV-1 to Vero cells (EC_50_ = 0.12 µg/mL, 0.38 µM) with a very high selectivity index (SI = 1728). Some in silico analysis performed with NQ **4** predicted that it could bind to the prefusion form of the E glycoprotein of DENV-2. In this context, our findings are a starting point to follow biopharmaceutical and pre-clinical toxicity and efficacy evaluations, focused towards the development and formulation of a pharmaceutical product that can prevent herpes and dengue infections avoiding the appearance of new drug-resistant strains.

Related to the potentiality of these quinones as antineoplastic, the anthraquinone AQ **11** was the most cytotoxic either on HeLa (IC_50_ = 0.01 µM) and Jurkat (IC_50_ = 1.4 µM) cell lines, with a low toxicity against Vero cells (IC_50_ = 321.9 µM) and therefore, with high SI, much better than the reference drug doxorubicin. These facts make AQ **11** a new, highly selective and promising lead compound, due not only to the experimental results found, but also to the favorable physicochemical properties and ADME (absorption, distribution, metabolism and excretion) data predictions, as to the lack of main toxicity risks. However, its drug likeness score (0.26) must be increased, and further virtual and experimental studies oriented to the development of a more consistent candidate to experimental preclinical toxicity and anticancer assays should be carried out.

## Figures and Tables

**Figure 1 molecules-24-01279-f001:**
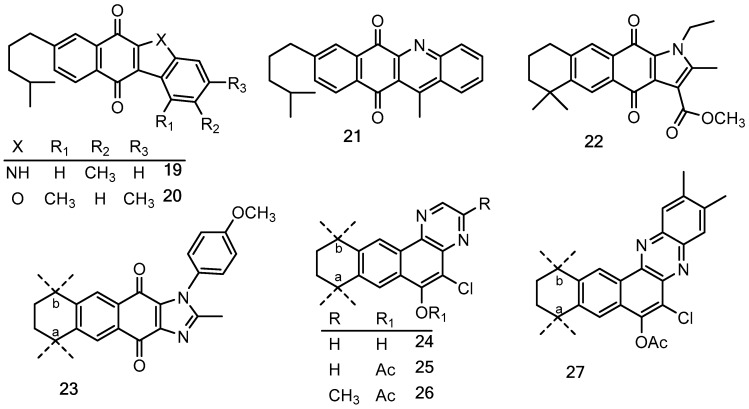
Chemical structures of the heterocycle-fused quinone derivatives (HetQs) tested. Compounds **23**–**27** are 1:1 mixtures of **a** and **b** regioisomers.

**Figure 2 molecules-24-01279-f002:**
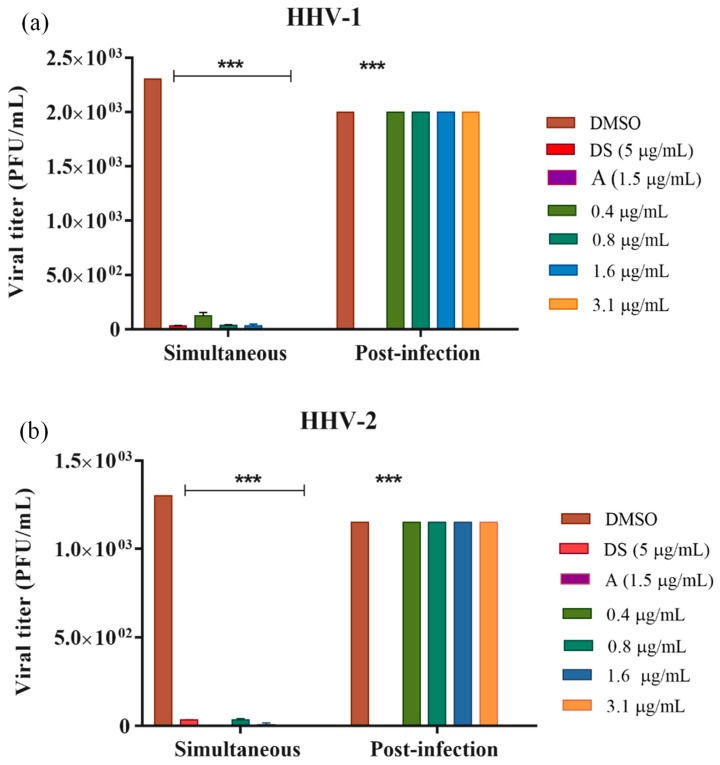
Effect of NQ **4** on the viral titer reduction of HHV-1, HHV-2 and DENV-2 in simultaneous and post-infection treatments. Vero or BHK-21 cells were infected with HHV-1 (**a**), HHV-2 (**b**) and DENV-2 (**c**) and treated with several concentrations of NQ **4** at different stages of the viral replicative cycle. Virions were quantified by plaque forming unit (PFU) titration (PFU/mL). Bars represent the mean ± SEM relative to DMSO control from two independent experiments. *p* values were determined by unpaired Student’s T-test (***, *p* < 0.001; **, *p* < 0.01). DMSO: dimethyl sulfoxide; DS: dextran sulfate (5 µg/mL); A: acyclovir, (1.5 µg/mL); H: heparin (10 µg/mL); R: ribavirin (3.7 µg/mL); NQ **4** against HHV serotypes (0.4–3.1 µg/mL); NQ **4** against DENV-2 (0.4–1.6 µg/mL).

**Figure 3 molecules-24-01279-f003:**
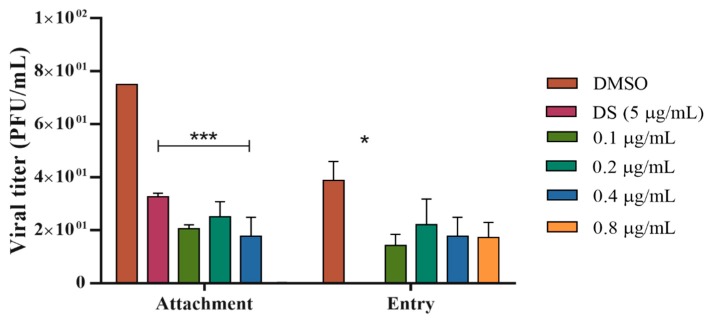
Effect of NQ **4** on HHV-1 viral attachment and entry on Vero cells. Vero cells were infected with HHV-1 and treated with different concentrations of NQ **4** at viral attachment or entry (see details in Materials and Methods). Bars represent the mean ± SEM relative to DMSO control from two independent experiments. *p* values were determined by unpaired Student’s T-test (***, *p* < 0.001; *, *p* < 0.05). DMSO: dimethyl sulfoxide; DS: dextran sulfate (5 µg/mL); NQ **4** (0.1–0.8 µg/mL).

**Figure 4 molecules-24-01279-f004:**
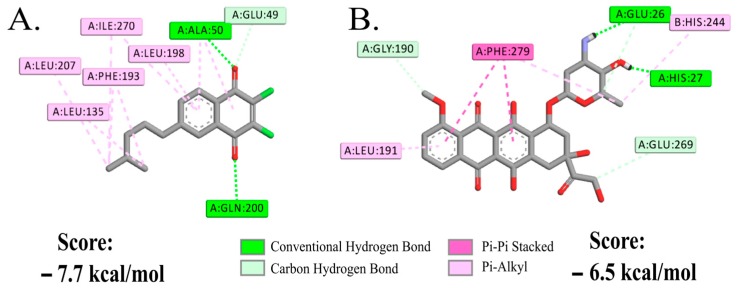
Interaction of NQ **4** and doxorubicin with DENV-2 prefusion envelope protein (ENV)**.** Interactions formed by NQ **4** (**A**) and doxorubicin (**B**) with amino acid residues reported for the hydrophobic detergent-binding pocket of the ENV protein (PDB:1OKE). Only hydrogen atoms involved in the interactions are depicted.

**Figure 5 molecules-24-01279-f005:**
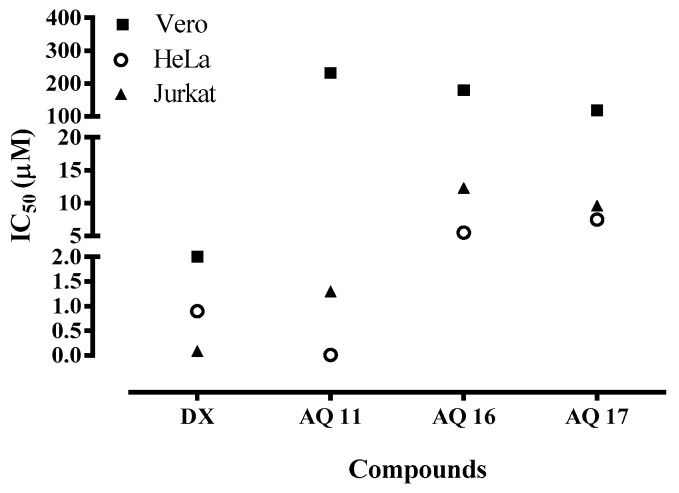
Relative cytotoxicity of the most selective quinones AQ **11**, AQ **16** and AQ **17** for HeLa and Jurkat tumor cells and for normal Vero cells, in comparison with the reference drug doxorubicin (DX).

**Table 1 molecules-24-01279-t001:** Structures of the terpenyl-1,4-naphthoquinone (NQ) and 1,4-anthraquinone (AQ) derivatives tested.

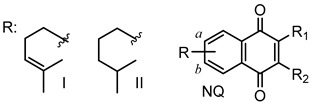				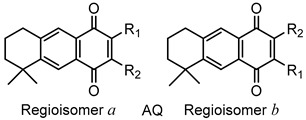		
**NQ** *(ratio a/b)*	**R**	**R_1_**	**R_2_**	**AQ** *(ratio a/b)*	**R_1_**	**R_2_**
**1**	I	H	H	**10** *(9:1)*	Cl	H
**2** *(9:1)*	I	Cl	H	**11** *(1:1)*	4-MeO-Ph-NH-	H
**3** *(1:1)*	II	Br	H	**12**	Cl	Cl
**4**	II	Cl	Cl	**13** *(9:1)*	AcNH-	Cl
**5**	II	Br	Br	**14** *(1:1)*	EtNH-	Cl
**6** *(9:1)*	II	Cl	4-MeO-Ph-NH-	**15** *(1:1)*	3,4-(Me)_2_-Ph-NH-	Cl
**7** *(1:1)*	II	Cl	4-MeO-Ph-O-	**16** *(1:1)*	4-MeO-Ph-NH-	Cl
**8** *(9:1)*	II	Br	4-MeO-Ph-NH-	**17** *(1:1)*	3,4-(MeO)_2_-Ph-NH-	Cl
**9** *(9:1)*	I	Cl	Ph-CH_2_-NH-	**18** *(1:9)*	3,4,5-(MeO)_3_-Ph-NH-	Cl

**Table 2 molecules-24-01279-t002:** Reduction of viral titer and antiviral activity against Human Herpesvirus type 1 (HHV-1) and 2 (HHV-2) on infected Vero cells of selected 1,4-naphthoquinones (NQ) and 1,4-anthraquinones (AQ).

Type	Compound	HHV-1		HHV-2	
		R*f* ^c^	1 TCID_50_ ^a^ (µg/mL) ^d^	R*f* ^c^	10 TCID_50_ ^b^ (µg/mL) ^d^
**NQ**	**2**	10^2^	6.25	10^1^	25
	**3**	*nd*	>50	10^2^	6.25
	**4**	**10^2^**	**6.25**	**10^2^**	**12.5**
	**6**	10^2^	6.25	*nd*	>50
	**7**	10^1^	50	*nd*	>50
	**8**	10^2^	25	*nd*	>50
**AQ**	**10**	10^2^	6.25	*nd*	>50
	**12**	*nd*	>50	10^2^	25
	**13**	*nd*	>50	10^1^	12.5
	**15**	10^1^	50	*nd*	>50
	**16**	10^1^	50	*nd*	>50
	**17**	10^1^	50	*nd*	>50
	**18**	10^1^	25	*nd*	>50
	**DS**	**10^2^**	**0.5**	*nd*	
	**A**	**10^4^**	**1.5**	**10^4^**	**1.5**

^a^ 1 TCID_50_: 1 Cell Culture Infectious Dose 50%; ^b^ 10 TCID_50_: 10 Cell Culture Infectious Dose 50%; ^c^
*Rf*: Reduction factor of the viral titer; ^d^ Non-toxic concentration that showed viral reduction factor; *nd*: Not determined; DS: Dextran sulfate; A: Acyclovir. The most potent compound is bolded for better comparison.

**Table 3 molecules-24-01279-t003:** Cytotoxicity results (IC_50_, µM) of 1,4-naphthoquinones (NQs), 1,4-anthraquinones (AQs) and heterocycle-fused quinones (HetQs) on HeLa, Jurkat and Vero cell lines after 48 h.

Type	Compd.	HeLa ATCC CRL-1958			Jurkat ATCC TIB-152			Vero ATCC CCL-81	
		IC_50_ ± SD	*r* ^2^	SI	IC_50_ ± SD	*r* ^2^	SI	IC_50_ ± SD	*r* ^2^
**NQs**	**1**	>104.0	*na*	*nd*	>104.0	*na*	*na*	*nd*	*nd*
	**2**	103.8 ± 3.8	0.9	1	**6.2 ± 0.2**	0.8	**15**	95.0 ± 2.1	0.9
	**3**	<19.5	*na*	>1	**1.6 ± 0.1**	0.7	8	12.8 ± 2.0	0.8
	**4**	46.6 ± 3.3	1.0	**14**	36.3 ± 2.0	0.9	**18**	**≥642.7**	*na*
	**5**	19.7 ± 0.3	1.0	1	23.5 ± 1.7	0.7	1	15.7 ± 2.7	0.8
	**6**	34.7 ± 3.2	0.8	2	47.5 ± 0.2	1.0	1	>62.8	*na*
	**7**	≤15.7	*na*	≥5	12.0 ± 1.8	0.8	6	73.2 ± 0.9	0.7
	**8**	≤14.1	*na*	≥9	≤56.5	*na*	≥2	**132.2 ± 11.2**	0.9
	**9**	32.6 ± 0.8	1.0	5	14.5 ± 0.5	1.0	10	**147.1 ± 9.8**	0.8
**AQs**	**10**	40.0 ± 1.1	0.9	10	48.0 ± 4.7	0.8	8	**397.9 ± 6.1**	0.9
	**11**	**0.010 ± 0.001**	1.0	**13967**	**1.4 ± 0.1**	1.0	**168**	**231.9 ± 6.9**	1.0
	**12**	38.5 ± 1.2	0.9	4	27.5 ± 0.6	0.9	5	**136.5 ± 4.4**	0.9
	**13**	>75.3	*na*	<1	**6.3 ± 0.3**	0.9	11	67.2 ± 1.3	0.7
	**14**	>78.7	*na*	*nd*	>78.7	*na*	*na*	*nd*	*nd*
	**15**	>63.5	*na*	<2	12.4 ± 0.4	1.0	10	**>126.9**	*na*
	**16**	**5.6 ± 0.4**	0.9	**32**	12.4 ± 0.4	1.0	**14**	**179.3 ± 12.9**	0.7
	**17**	**7.5 ± 0.6**	0.9	**16**	**9.6 ± 0.4**	0.9	12	**118.6 ± 15.9**	0.8
	**18**	≤13.7	*na*	≥2	10.1 ± 0.8	0.9	2	24.3 ± 3.8	0.9
**HetQs**	**19**	77.0 ± 1.9	1.0	3	>72.4	*na*	<3	**212.5 ± 8.6**	0.7
	**20**	>69.3	*na*	<2	42.2 ± 0.7	0.7	4	**153.1 ± 4.5**	1.0
	**21**	>69.9	*na*	*nd*	>69.9	*na*	*na*	*nd*	*nd*
	**22**	73.0 ± 1.6	1.0	3	71.7 ± 1.8	1.0	3	**183.1 ± 15.9**	0.7
	**23**	15.5 ± 1.1	0.8	2	13.5 ± 0.7	0.9	3	36.0 ± 1.5	0.9
	**24**	55.9 ± 2.0	0.9	5	24.3 ± 1.5	0.9	12	**282.0 ± 28.2**	0.5
	**25**	>70.5	*na*	<4	13.8 ± 0.4	1.0	**20**	**269.5 ± 5.7**	1.0
	**26**	77.5 ± 3.6	0.9	2	43.6 ± 1.2	1.0	4	**182.7 ± 16.3**	0.7
	**27**	57.1 ± 2.2	1.0	3	16.6 ± 0.9	0.9	12	**197.3 ± 25.7**	0.6
	**Doxorubicin**	**0.9 ± 0.1**	0.9	2.2	**0.10 ± 0.01**	0.7	**22**	2.0±0.1	0.8

HeLa: Human cervix epithelial carcinoma, ATCC CRL-1958; Jurkat: human acute T cell leukemia, ATCC TIB-152; Vero: *Cercopithecus aethiops*, African green monkey kidney cell line, ATCC CCL 81; SD: Standard deviation; ***r*^2^**: Linear regression coefficient; SI: Selectivity Index (IC_50_ Vero/IC_50_ tumor cell (HeLa or Jurkat)); *na*: Not applicable; *nd*: Not determined. The IC_50_ values corresponding to the most cytotoxic compounds for cancer cells (<10 μM) or less toxic for normal cells (>100 μM), as well as SI values >14, are in bold for better comparisons.

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
