# Peer review of "Anti-Herpetic, Anti-Dengue and Antineoplastic Activities of Simple and Heterocycle-Fused Derivatives of Terpenyl-1,4-Naphthoquinone and 1,4-Anthraquinone"

_molecules, 2019, doi:10.3390/molecules24071279_

Round 1

Reviewer 1 Report

Comments

The manuscript by Castro and co-workers concerns the synthesis of Quinones and its derivatives using Diels-Alder cycloaddition of myrcene and p-benzoquinones followed by appropriate transformations. Studied the biological activity of these novel compounds on varies diseases like anti-herpetic, anti-dengue and antineoplastic. Overall the manuscript is written well, and the results are impressive, therefore, the present manuscript is suitable for molecules after considering the following modifications.

1)    The conclusion part should be rewritten because it is very generalized.

2)    The references are not properly written, there are sever mistakes in punctuations, (Ex: J Am         Chem Soc should be J. Am. Chem. Soc)

Author Response

The manuscript by Castro and co-workers concerns the synthesis of Quinones and its derivatives using Diels-Alder cycloaddition of myrcene and p-benzoquinones followed by appropriate transformations. Studied the biological activity of these novel compounds on varies diseases like anti-herpetic, anti-dengue and antineoplastic. Overall the manuscript is written well, and the results are impressive, therefore, the present manuscript is suitable for molecules after considering the following modifications.

We thank these positive comments

1)    The conclusion part should be rewritten because it is very generalized.

The conclusion section has now rewritten in a more specific manner.

2)    The references are not properly written, there are sever mistakes in punctuations, (Ex: J Am         Chem Soc should be J. Am. Chem. Soc).

All the references have been revised and the mistakes in punctuations and abbreviations have been corrected.

Reviewer 2 Report

This manuscript is published in the journal compatibly.

The authors showed the quinone skeleton compounds to play inhibitions for various human viruses. The research results were based on the wet and dry lab studies as well as gave some useful candidates for drug development.

Author Response

The authors showed the quinone skeleton compounds to play inhibitions for various human viruses. The research results were based on the wet and dry lab studies as well as gave some useful candidates for drug development.

We thank the reviewer for this comment

(x) Moderate English changes required

The English language and style have been carefully revised throughout the manuscript

Reviewer 3 Report

This manuscript describes anti-herpetic, anti-dengue, and antineoplastic activities of NQ, AQ, and HetQ.  I think it is well written and include interesting results.  Therefore, I think this manuscript is acceptable as an original article in Molecules.  However, some English errors are seen.  I recommend the authors to check the manuscript carefully.

Author Response

This manuscript describes anti-herpetic, anti-dengue, and antineoplastic activities of NQ, AQ, and HetQ.  I think it is well written and include interesting results.  Therefore, I think this manuscript is acceptable as an original article in Molecules.  However, some English errors are seen.  I recommend the authors to check the manuscript carefully.

Thank you for this comment

(x) English language and style are fine/minor spell check required

The English language and style have been carefully revised throughout the manuscript